# *Rhododendron luteum* Sweet Flower Supercritical CO_2_ Extracts: Terpenes Composition, Pro-Inflammatory Enzymes Inhibition and Antioxidant Activity

**DOI:** 10.3390/ijms25189952

**Published:** 2024-09-15

**Authors:** Lena Łyko, Marta Olech, Urszula Gawlik, Agnieszka Krajewska, Danuta Kalemba, Katarzyna Tyśkiewicz, Narcyz Piórecki, Andriy Prokopiv, Renata Nowak

**Affiliations:** 1Department of Pharmaceutical Botany, Medical University of Lublin, ul. Chodźki 1, 20-093 Lublin, Poland; 2Department of Biochemistry and Food Chemistry, University of Life Sciences, ul. Skromna 8, 20-704 Lublin, Poland; 3Institute of Natural Products and Cosmetics, Lodz University of Technology, ul. Stefanowskiego 4/10, 90-924 Łódź, Poland; 4Supercritical Extraction Department, Łukasiewicz Research Network-New Chemical Syntheses Institute, ul. Tysiąclecia Państwa Polskiego 13a, 24-110 Puławy, Poland; 5Bolestraszyce Arboretum and Institute of Physiography, Bolestraszyce 130, 37-722 Wyszatyce, Poland; 6Institute of Physical Culture Sciences, Medical College, University of Rzeszow, ul. Cicha 2A, 35-326 Rzeszow, Poland; 7Department of Botany, Botanical Garden, Ivan Franko National University of Lviv, 79005 Lviv, Ukraine

**Keywords:** *Rhododendron luteum*, Ericaceae, terpenoids, antioxidant, anti-inflammatory, liquid chromatography with mass spectrometry, supercritical CO_2_ extraction, headspace solid-phase microextraction, gas chromatography

## Abstract

Terpenes are plant secondary metabolites known for their anti-inflammatory and antioxidant activities. According to ethnobotanical knowledge, *Rhododendron luteum* Sweet was used in traditional medicine against inflammation. The present study was conducted to determine the triterpene profile and antioxidant and anti-inflammatory activity of supercritical CO_2_ (SC-CO_2_) extracts of *Rhododendron luteum* Sweet flower (RLF). An LC-APCI-MS/MS analysis showed the presence of eight pentacyclic triterpenes and one phytosterol in the extracts obtained with pure CO_2_ as well as CO_2_ with the addition of aqueous ethanol as a co-solvent. Among the compounds detected, oleanolic/ursolic acid, β-sitosterol and 3*β*-taraxerol were the most abundant. The extract obtained with pure SC-CO_2_ was additionally subjected to HS-SPME-GC-FID-MS, which revealed more than 100 volatiles, mainly eugenol, *β*-phenylethanol, dodecane, *β*-caryophyllene, estragole and (*Z*)- and (*E*)-cinnamyl alcohol, followed by *δ*-cadinene. The extracts demonstrated significant hyaluronidase inhibition and exhibited varying modes of lipoxygenase and xanthine oxidase inhibitory activities. The studies of RLF have shown that their SC-CO_2_ extracts can be a rich source of triterpenes with anti-inflammatory potential.

## 1. Introduction

Inflammation is the primary mechanism for tissue repair after injury or infection. An inflammatory response implicates a cascade of biochemical events driven by a set of mediators. Uncontrolled inflammation plays a key role in the pathogenesis of many chronic diseases, including rheumatoid arthritis, osteoarthritis, inflammatory bowel disease, chronic arterial and venous disease, myocardial ischemia, aerial hypertension, and Alzheimer’s disease [1,2]. Moreover, cancers can also arise from the inflammation sites [3]. Treatment of inflammation with available drugs has a high risk of side effects, including cardiovascular, renal and gastrointestinal toxicity [4]. Thus, searching for new natural remedies could be a promising alternative.

The *Rhododendron* genus belonging to the Ericaceae family comprises over 1000 species covering most of the Northern Hemisphere. Phytochemical studies on the genus established the presence of many biologically active secondary metabolites, including flavonoids, coumarins, terpenes and sterols [5,6]. Many of them were isolated and subjected to biological studies in which they exerted various kinds of pharmacological activities. Among them, anti-inflammatory, antioxidant, anti-HIV, antidiabetic and anticancer potential were observed [5,7,8].

One of the *Rhododendron* representatives is *R. luteum* Sweet (yellow azalea or honeysuckle azalea; RL). Investigations of the specimen growing in Turkey revealed selective anticancer potential, antibacterial activity and inhibitory activity against *α*-glucosidase and tyrosinase [9,10,11]. Our previous studies on Polish specimens were focused on the polyphenolic composition of RL leaf samples. Different types of extracts and highly concentrated phenolic fractions from *R. luteum* leaf were subjected to phytochemical analyses as well as anti-inflammatory and antioxidant tests. It was proven that RL leaf extracts and polyphenols possess the ability to inhibit enzymes involved in inflammation [12,13]. Some previous studies reported that other parts of other species of rhododendrons are also sources of highly active molecules belonging to different chemical groups, e.g., triterpenes, diterpenes, volatiles, flavonoids, lignans and sterols [14,15,16]. However, the knowledge about the content of these secondary metabolites and their influence on RL biological activity remains very limited. Notably, the chemical profile of RL flowers is poorly examined, where studies of flowers collected from Eastern European specimens have not been reported to date.

Among phytochemicals that act as potential anti-inflammatory agents are terpenes. They include monoterpenes, diterpenes, sesquiterpenes, triterpenes and tetraterpenes. Terpenoids are also types of terpenes containing oxygen in molecules, and they occur as alcohols, aldehydes, esters, ethers, epoxides, ketones and phenols. A number of in vitro studies indicate that terpenes affect pro-inflammatory mediator signaling pathways. It was observed that some of them could suppress NO release and reduce the level of TNF-α, interleukins, and prostaglandin PGE_2_ [4,17]. Among the many mechanisms of anti-inflammatory action, some terpenes could also block the enzymes implicated in inflammatory processes such as cyclooxygenase-2, hyaluronidase, lipoxygenase, and xanthine oxidase [18,19,20,21].

Non-polar and weakly polar compounds (e.g., triterpenes, sterols and volatiles) can be extracted from plant material by various classic and modern techniques. Supercritical fluid extraction (SFE), belonging to the modern techniques, is an excellent tool for the efficient isolation of compounds with low polarity [22]. Compared with traditional extraction methods, SFE allows for the extraction of thermolabile compounds due to near-ambient critical temperature. Moreover, it provides higher selectivity while reducing extraction time. Another advantage of this method is the use of carbon dioxide as a solvent. It is non-toxic, easily available and entirely removable from an extract [22,23]. On the other hand, SFE may be much more complex in handling and optimization as fine-tuning parameters such as pressure, temperature and particle size is more challenging. Moreover, it often requires a higher initial investment compared with traditional extraction techniques. This cost factor is due to the need for specialized equipment to handle the high pressures and temperatures involved [24]. However, if the apparatus itself is available, the cost of carbon dioxide is low. SFE processing was successfully used for the highly efficient recovery of active metabolites from different natural matrices [25,26]. However, to the best of our knowledge, supercritical CO_2_ extracts prepared from RL have not been studied to date.

Because the content of non-polar and weakly polar secondary metabolites in RL flowers and their pharmacological potential has not been reported, the present study was undertaken to investigate the chemical profile and anti-inflammatory potential of two types of *R. luteum* supercritical CO_2_ (SC-CO_2_) extracts. The content of triterpenes and sterols in each SC-CO_2_ sample was estimated by liquid chromatography/atmospheric pressure chemical ionization-triple quadrupole mass spectrometry (LC-APCI-MS/MS). Moreover, volatiles in pure CO_2_ extracts were identified by gas chromatography with mass spectrometry and flame ionization detection (HS-SPME-GC-FID-MS). The potential to inhibit the enzymes involved in the inflammatory process, such as lipoxygenase and xanthine oxidase, was examined. Additionally, the hyaluronidase inhibition ability and oxygen radical absorbance capacity of SC-CO_2_ extracts were investigated.

## 2. Results and Discussion

### 2.1. Extraction Yield and Recovery of Triterpenes

Our study is the first one presenting the profile of triterpenes and sterols of *R. luteum* flowers. For the efficient extraction of these mostly non-polar and weakly polar compounds, supercritical CO_2_ (SC-CO_2_) extraction was chosen. Because previous studies have reported that RLF might also contain some more polar molecules, e.g., pentacyclic triterpene acids, it was decided to include second stage of SFE (Figure 1) [27]. In the first step of the SFE, the plant material was extracted with pure CO_2_ to obtain a high content of non-polar terpenes and phytosterols. In the next step, polar co-solvents (30% aqueous ethanol) were subsequently used to elute more polar compounds [22]. This exhaustive procedure allowed us to increase the yield and expand the range of polarity of the terpene profile extracted from the plant material (see the extraction efficiencies in Figure 1). The composition of both RLF extracts was investigated with a newly created LC-APCI-MS/MS method. It is worth noting that the analysis was developed to track several compounds belonging to different chemical classes in order to determine the content of triterpenes and phytosterol in a single run (Appendix A).

As a result, pentacyclic triterpenes belonging to olean-type (oleanolic acid, maslinic acid, 3*β*-taraxerol and erythrodiol), ursolic-type (ursolic acid, *α*-amyrin, corosolic acid, euscaphic acid and uvaol) and lupane-type (lupeol), as well as *β*-sitosterol belonging to phytosterols, were detected and quantified (Table 1, Figure 2).

In the extract of RLF-CO_2_ (obtained with the use of CO_2_ without any co-solvents), there was a large amount of non-polar compounds, i.e., sterol and pentacyclic triterpenoids. One of them, *β*-sitosterol (80.10 ± 0.42 mg/g of dry extract (DE)), was previously observed in the SFE extract of another genus representatives, *R. adamsii* and *R. sichotense*. [28]. According to Sajfrtová et al. [29], a temperature of 40 °C is optimal for the extraction of *β*-sitosterol, although the extraction pressure would have to be reduced to 150 bar to obtain a higher concentration of this compound.

The compound detected in the second highest amount in this extract was 3*β*-taraxerol (49.39 ± 0.05 mg/g of DE). This molecule is known for its multiple pharmacological properties, e.g., anti-inflammatory, anti-fungal, antimicrobial and anti-diabetic properties [30]. Taraxerol was previously identified in chloroform extracts of *R. luteum* as well as *R. schlippenbachii* and *R. dauricum* extracts [31]. It was also reported to occur in *R. ovatum* and *R. molle* [5]. Other pentacyclic triterpenes observed in RLF-CO_2_ extract were lupeol (17.00 ± 0.14 mg/g of DE), *α*-amyrin (28.45 ± 0.07 mg/g of DE) and erythrodiol/uvaol (3.60 ± 0.03 mg/g of DE), which were also detected in other *Rhododendron* species, i.e., *R. tomentosum*, *R. dauricum*, *R. adamsii*, *R. colletianum*, *R. brachycarpum* and *R. formosanum* [5,15,28,32]. More polar triterpenes such as oleanolic/ursolic and maslinic acid were found in significantly lower amounts when corosolic and euscaphic acids occurred in trace amounts. Oleanolic and ursolic acid were previously reported in *Rhododendron luteum* Sweet leaves, and their quantitative ratio was 3:7, respectively [27,33]. These acids have a very similar structure, *m*/*z* of the precursor ion and fragmentation pattern. Despite testing numerous gradients, mobile phase modifications and temperatures, we were not able to reach a satisfying separation of these compounds on the Kinetex XB-C18 column. We observed both compounds in our samples. However, the difference in their retention times was small, and we decided that it would be more reliable to provide a sum of ursolic and oleanolic acid (based on the calibration curve prepared for oleanolic acid). Similarly, this was the case for erythrodiol and uvaol. Their separation was not achieved. Therefore, their quantitation was performed using a calibration curve prepared for erythrodiol, and the results were expressed as erythrodiol. The same separation issue was previously reported by other researchers [26,34,35].

As shown in Table 1, the addition of polar co-solvent resulted in obtaining the samples with a predominant content of more polar triterpenes. The use of ethanol in the second step of the SFE increased the extraction efficiency in comparison with the first step (4.91% in RLF-CO_2_, 10.24% in RLF-CO_2_ + 30%EtOH), which is comparable with results obtained in another study [36]. In the cited paper, the addition of ethanol as a co-solvent highly increased extraction yield. However, in RLF-CO_2_ + 30%EtOH, a significantly lower content of some triterpenes with lower polarity was observed (Table 1). According to Wrona et al. [22], the parameters used in the second step of our SFE are optimal for the extraction of phenolic compounds. Simultaneously, the addition of aqueous ethanol greatly increased the content of more polar triterpene acids such as oleanolic and ursolic acids (94.35 ± 0.35 mg/g of DE), which is in agreement with some previous observations [37]. According to the authors of the cited paper, the addition of aqueous water and ethanol highly improves the extraction yield and enhanced solubility of oleanolic and ursolic acids after increasing the polarity of the solvent. These triterpenoid acids were previously found in other *Rhododendron* species. They were isolated together from ethanolic extracts of *R. dauricum* [15]. Similarly, high amounts of oleanolic and ursolic acids were observed in other *Ericaceae* representatives. For example, the SFE extract of *Calluna vulgaris* obtained with the addition of 15% ethanol contained a comparable amount of oleanolic and ursolic acid with our RLF-CO_2_ + 30%EtOH [36]. In turn, in *Gaultheria procumbens* L. (Ericaceae), these compounds accounted for almost 40% of the extract [38]. Another oleane derivative, maslinic acid, was isolated from *R. collettianum,* and it was found in *R. anthopogonoides* [5,39]. Oleanolic and ursolic acid are triterpene acids commonly found in plants, including fruit and vegetables. They are recognized as non-toxic and safe to consume [40,41].

This study presents the first comprehensive analysis of the triterpene profile of *Rhododendron luteum* Sweet flowers. Our findings reveal the presence of compounds such as erythrodiol, uvaol, *β*-sitosterol, *α*-amyrin, corosolic, maslinic and euscaphic acids, which were not previously detected in any RL material.

The proposed two-step supercritical extraction scheme is an effective, environmentally friendly and efficient method for recovering terpenoid compounds of varying polarity and volatility from raw materials. An additional advantage of this method is the ability to conduct the process at low temperatures and obtain a concentrated extract with high microbiological purity, in accordance with the principles of green chemistry.

### 2.2. Content of Volatile Compounds in RL Samples

The HS-SPME-GC-FID-MS analysis of RLF-CO_2_ revealed more than 100 compounds, presented in Table 2. Different types of constituents were found in the analyzed sample. The group of identified volatile compounds that was most numerous were terpenes: mono- and sesquiterpene hydrocarbons and their oxygenated derivatives. Other important groups were aliphatic compounds and compounds with an aromatic benzene ring.

The main volatile constituent in RLF-CO_2_ extract (see Figure 3) was found to be benzyl alcohol (10.65%). Eugenol (7.26%) and estragole (5.80%), which were also detected in large amounts, show a range of biological activities, including anti-inflammatory and antiproliferative activities [43,44,45]. The other observed volatiles were: *β*-phenylethanol (5.20%), dodecane (5.07%), *δ*-cadinene (3.75%), *β*-caryophyllene (3.09%), (*E*)- and (*Z*)-cinnamyl alcohol (3.77% and 2.05%) and dihydroactinidiolide (1.94%). The mechanisms of the anti-inflammatory activity of these compounds are shown in Table 3. Some volatiles also present in SC-CO_2_ extracts in large amounts, such as δ-cadinene, have not yet been well studied, although they are components of essential oils known for their anti-inflammatory properties [46,47]. Others, such as dodecane itself, have no reports of anti-inflammatory activity. However, dodecane is often used as a solvent or carrier, e.g., in formulations such as cosmetics [48]. On the basis of common mass ion 222 and the similar ion pattern of mass spectra, the two unidentified compounds (4.23% and 1.88%) were probably isomeric sesquiterpene alcohols.

Previously, only a few reports on the volatiles of *R. luteum* were published. Different methods were used for the volatiles isolation of *R. luteum* collected in Turkey [11,42,43]. The most comparable with our research is the paper of Tasdemir et al. [42], who analyzed the headspace of hexane and dichloromethane extracts of different *Rhododendron* species. Two alcohols with a benzene ring that belonged to the main compound in our research, namely benzyl alcohol and *β*-phenyl-ethanol, were also among the main volatiles in the dichloromethane extract of *R. luteum* flowers (16.6% and 4.3%, respectively) followed by limonene, p-cymene and 1,8-cineole. On the contrary, previously the hydrocarbons *β*-caryophyllene (34.0%), *α*-pinene (10.0%) and (*E*)-*β*-ocimene (10.4%), as well as methyl benzoate (11.7%), were the main constituents in the headspace of aerial plant parts [43], and the terpene alcohols *α*-terpineol, T-muurolol and *α*-cadinol, as well as benzyl salicylate were dominant and no monoterpene hydrocarbons were identified in essential oil of *R. luteum* flowers [11].

### 2.3. Biological Activity of RLF Extracts

Activities of enzymes such as hyaluronidase, lipoxygenase (LOX) or xanthine oxidase (XO) are implicated in a range of pathophysiological processes, including chronic inflammation, atherosclerosis and cancers [56,57,58]. Selective inhibition of those enzymes may provide a strategy for the alleviation of inflammatory disorders and minimize side effects at the same time. Hence, it was decided to examine the ability of RLF SC-CO_2_ samples to inhibit hyaluronidase, LOX and XO.

Hyaluronic acid is involved in tissue regeneration, angiogenesis and inflammation response. Hyaluronidase activity leads to the degradation of hyaluronic acid in the extracellular matrix and increases the permeability of connective tissues during inflammation [56]. Moreover, the product of its activity stimulates the release of proinflammatory cytokines and expression of inflammation-related genes and increases immunity. Therefore, a substance with the ability to block hyaluronidase could potentially act as an anti-inflammatory agent [59].

As presented in Table 4, IC_50_ values of the samples vary depending on the extract type. The most potent hyaluronidase inhibitor was found to be RLF-CO_2_ + 30%EtOH (23.75 ± 1.44 μg of DE/mL), possessing an intermediate content of the tested triterpenes. Sample RLF-CO_2_ (IC_50_ = 106.98 ± 4.45 μg of DE/mL) presented a significantly lower potential. Probably, the activity of RLF-CO_2_ + 30%EtOH is affected by most polar terpenoids and also by other types of eluted compounds, e.g., polyphenols. It was previously observed that the addition of ethanol during SFE has a significant influence on the elution of phenolic compounds [60]. Previous studies suggested the relevant potential of triterpenes as enzyme inhibitors [61,62]. However, our study shows low OA activity compared with that of samples. Therefore, it can be suspected that the anti-hyaluronidase activity of RLF-CO_2_ + 30%EtOH may be greatly influenced not only by triterpenes and sterols but also by other plant secondary metabolites [63].

Lipoxygenases catalyze the dioxygenation of polyunsaturated fatty acids, producing biologically active eicosanoids. They were proven to regulate aspects of cancer development and play a key role in inflammation. Lipoxygenase-catalyzed linoleic and arachidonic acids potentially affect cell growth, angiogenesis and cell invasion [57,64]. Thus, the next step of the study was the determination of whether the extracts present the ability to inhibit lipoxygenase and additionally, the identification of the mode of their action.

According to the results presented in Table 4, both extracts demonstrated similar anti-LOX activity. The IC_50_ values were 0.50 ± 0.02 and 0.67 ± 0.03 mg of DE/mL for RLF-CO_2_ + 30%EtOH and RLF-CO_2_, respectively. Higher activity was demonstrated by the sample RLF-CO_2_ + 30%EtOH prepared with the use of modifier (aqueous EtOH). Oleanolic acid (OA) was tested as a triterpene reference. It presented a similar IC_50_ value of 0.68 ± 0.03 mg of DE/mL. In turn, quercetin, known for its ability to inhibit LOX, presented significantly lower activity. It suggests that terpenes may contribute to the high anti-LOX activity of RLF extracts. Among the constituents detected in RLF-CO_2_, β-caryophyllene, α-pinene and limonene were previously reported to exert inhibitory effects on LOX. [65] Cinnamyl alcohol, belonging to the main volatile constituents in RLF-CO_2_, showed anti-LOX activity in another study (IC50 value of 250 µM) [54]. Also, β-sitosterol and lupeol found in RLF-CO_2_ could effectively inhibit LOX in some previous studies [38,66]. Concerning polar extracts, their main constituents, oleanolic and ursolic acids, were also previously shown to inhibit the activity of LOX. [38]. In the next step, the inhibition mode of RL samples was defined with use of Lineweaver–Burk plot analysis. As presented in Figure 4, all of the tested samples, as well as oleanolic acid, presented a competitive mechanism of anti-LOX action [67,68]. Also, Werz [69] highlighted the role of sesquiterpenoids and pentacyclic terpenoids in LOX inhibition through interference with the substrate-binding site of the enzyme. However, another study suggested the presence of a synergistic inhibitory effect of terpenes and flavonoids in plant extracts [18]. Therefore, the final anti-LOX effect of RL samples is probably affected by the various groups of the SFE-eluted constituents.

Another enzyme implicated in the inflammatory response is xanthine oxidase (XO), which catalyzes the hydroxylation of xanthine to uric acid. Its excessive production or inadequate excretion may result in gout. [58] Hence, it was decided to perform an anti-XO assay. The RLF SC-CO2 samples were found to be moderate inhibitors of xanthine oxidase. RLF-CO_2_ + 30%EtOH presented similar activity to the reference compound OA (2.36 ± 0.09 mg of DE/mL and 2.08 ± 0.09 mg of DE/mL, respectively). The RLF-CO_2_ sample demonstrated a twice as low potential. The activity of RL SFE extracts is probably related largely to the high content of triterpenes. Some of the identified constituents (e.g., taraxerol) were reported as effective XO inhibitors (IC_50_ value 12.87 ± 5.10 μg/mL) [70]. The higher activity of RLF-CO_2_ + 30%EtOH may be due to the presence of oleanolic/ursolic acid [71]. Interestingly, the extracts, which are mixtures of triterpenes and other active compounds, present an uncompetitive (mixed) mode of XO inhibition, while the solution of pure oleanolic acid demonstrate a non-competitive mode of action (Figure 5) [67,68].

Free radicals may injure cells by the oxidative degradation of different cellular components or via altering the balance between protease and anti-protease, leading to inflammation [72]. Unlikely popular antioxidant assays following a single electron transfer mechanism, the ORAC test evaluates antioxidant activity, taking into account the kinetics of the reaction [73]. Hence, it was decided to examine the antioxidant activity of RLF SC samples in ORAC assay. As can be seen in Table 3, the RLF-CO_2_ + 30%EtOH sample presented an antioxidant potential of 338.99 ± 8.6 mg of Trolox/g of DE. The activity of RLF-CO_2_ was three times lower (119.92 ± 7.336 mg of Trolox/g of DE), which may be associated with the lower content of more polar components, e.g., triterpenic acids. However, components contained in examined extracts (oleanolic acid, uvaol and erythrodiol) tested separately showed no activity in the ORAC assay [74]. RL SC flower extracts were found to be significantly less active antioxidant agents than *Rhododendron luteum* leaf methanolic extracts and their polyphenolic-rich fractions [12,13]. Therefore, it is possible that the higher oxygen radical scavenging potential of RLF-CO_2_ + 30%EtOH may also be connected with the presence of some polyphenols eluted with the use of co-solvent (water and ethanol, 70:30, *v*/*v*). Triterpene-rich supercritical CO_2_ extracts can be a natural source of efficient antioxidant molecules [26]. However, it can be noticed that triterpenes show relatively weak antioxidant potential in the ORAC assay. The Trolox equivalent determined for the oleanolic acid was found to be 54.39 ± 2.81 mg/g of DE. This observation is in agreement with some previous studies [74,75]. Therefore, it can be assumed that other metabolites, e.g., phenolic compounds, may have considerable influence on the antiradical activity of RL samples.

## 3. Materials and Methods

### 3.1. Plant Material

Flowers of *Rhododendron luteum* Sweet were collected in the Botanical Garden of Bolestraszyce (Poland) in May 2022 and immediately frozen. The voucher specimen was deposited at the Department of Pharmaceutical Botany, Medical University of Lublin, Poland (voucher No. RL-02/22). The plant material was lyophilized, powdered, and sifted through a sieve (aperture dimension: ø1.5 mm). The ground samples were then vacuum-packed and stored at 4 °C until further use.

### 3.2. Chemicals and Apparatus

Analytical standards of *α*-amyrin, maslinic acid, oleanolic acid, *β*-sitosterol and stigmasterol, as well as LC-MS-grade methanol, acetonitrile, hyaluronidase, Trolox (6-hydroxy-2,5,7,8-tetramethylchroman-2-carboxylic acid), 2-methylpropionamide dihydrochloride (AAPH), ferrozine (3-(2-pyridyl)-5,6-bis-(4-phenyl-sulfonic acid)-1,2,4-triazine), soybean 15-lipooxygenase, linoleic acid, quercetin, xanthine oxidase, allopurinol, bovine serum albumin, hyaluronic acid, sodium phosphate monobasic solution, sodium phosphate dibasic solution, sodium chloride solution, sodium acetate, and acetic acid were purchased from Sigma-Aldrich Fine Chemicals (St. Louis, MO, USA). Lupeol, ursolic acid, betulin and betulinic acid were from ChromaDex (Irvine, CA, USA). Chloroform was supplied by Avantor Performance Materials Poland S.A. (Gliwice, Poland). For obtaining LC-MS-grade water, a Millipore Direct-Q3 purification system (Bedford, MA, USA) was used. For the evaporation of all samples, a Heidolph Basis Hei-VAP Value evaporator (Schwabach, Germany) was used. Lyophilization was conducted in a Free Zone 1 apparatus (Labconco, Kansas City, KS, USA). Spectrophotometric measurements were conducted with the use of an Infinite Pro 200F microplate reader from Tecan Group Ltd. (Männedorf, Switzerland) and transparent or black 96-well microplates (Nunclon, Nunc; Roskilde, Denmark).

### 3.3. Supercritical Fluid Extraction (SFE)

The supercritical fluid extraction of *R. luteum* flowers was conducted using a laboratory-scale installation for SFE with supercritical carbon dioxide (SC-CO_2_), with the dynamic supercritical fluid extractor being operated at temperatures up to 80 °C and a pressure of up to 450 bar. The extraction process was divided into two stages, as presented in Figure 1. Plant material (100 g) was firstly extracted with pure carbon dioxide without the addition of co-solvent at 40 °C and 300 bar for 60 min. In the second phase, the residue from the first stage was extracted with the addition of 30% aqueous ethanol (C_2_ H_5_ OH:H_2_ O, 70:30, *v*/*v*) as a co-solvent at 50 °C and 300 bar for 120 min. The temperature increase in the second stage was applied to maintain supercritical conditions. The average SC-CO_2_ flow rate was 10 kg of CO_2_/h. Each extract was evaporated, lyophilized and weighed. The extraction yield was calculated according to the following equation:(mEX/mPM) × 100%,(1)
where mEX is the mass of dry extract obtained in each particular step and mPM is the mass of the dry plant material.

### 3.4. LC-APCI-MS/MS Analysis of Triterpenes

Samples were injected into an Agilent 1200 LC system (Agilent Technologies, Santa Clara, CA, USA) and separated on a Kinetex XB-C18 column (150 × 2.1 mm; particle size 2.6 µm, Phenomenex, Torrance, CA, USA). A gradient elution with water with 0.1% formic acid (solvent A) and ACN with 0.1% formic acid (solvent B) at 35 °C was used (gradient details are given in Appendix A). The flow rate was 350 µL/min, and the injection volume was 5 µL. The LC was connected a 3200 QTRAP triple quadrupole mass spectrometer equipped with a Turbo V™ source (Sciex, Redwood City, CA, USA). The LC-MS system was controlled by Analyst 1.5 software (Sciex, Redwood City, CA, USA). An atmospheric pressure chemical ionization (APCI) was applied using both positive and negative modes. The instrument parameters for APCI (−) were as follows: the collision gas was set at 3, nebulizer current at −5, curtain gas at 20 psi, temperature at 450 °C and ion source gas at 25 psi. For APCI (+), the collision gas was set at 4, nebulizer current at 4, curtain gas at 30 psi, temperature at 300 °C and ion source gas at 35 psi. The detection and quantification of analytes were performed using the multiple reaction monitoring (MRM) mode. The most intense MRM transitions and their optimal parameters were determined experimentally for each compound and are provided in the Appendix A. The LOD (limit of detection, at a signal-to-noise ratio of 5:1), LOQ (limit of quantification, at a signal-to-noise ratio of 10:1) and calibration curves were established using the corresponding standard solutions (Appendix A). All LC-MS analyses were performed at least three times for each standard compound and sample.

### 3.5. Analysis of Volatile Compounds

For the analysis of volatile compounds from *R. luteum* flower SFE extracts, the HS-SPME-GC-FID-MS technique was used. Sampling was carried out using SPME grey fiber 50/30 μm DVB/CAR/PDMS, StableFlex 2 cm (Supelco, Bellefonte, PA, USA). An extract (25 mg) of SFE was introduced to a 15 mL Amber vial with a PTFE/Silicone Septa hole cap (Supelco Bellefonte, USA). Before extraction, each sample was incubated at 60 °C for 30 min. Then, the fiber was introduced into the vial, and a 30 min absorption was performed. The fiber was directly desorbed in GC-FID-MS for 10 min. The analysis was performed with two repetitions. For the analysis, a Trace GC ultra gas chromatograph coupled with a DSQ II mass spectrometer (Thermo Electron Corporation; Waltham, MA, USA) was used. A simultaneous GC-FID and MS analysis was performed using an MS-FID splitter (SGE, Analytical Science, Austin, TX, USA). The operating conditions were: non-polar capillary column Rtx-1ms (60 m × 0.25 mm, 0.25 m film thickness); programmed temperature: 50 (3 min)–300 °C, 4 °C/min; injector (SSL) temperature: 280 °C; detector (FID) temperature: 300 °C; transfer line temperature: 250 °C; carrier gas: helium; flow with constant pressure: 200 kPa; and split ratio: 1:20. The mass spectrometer parameters were: ion source temperature: 200 °C; ionization energy: 70 eV (EI); scan mode: full scan; and mass range: 33–420. The percentages of constituents were computed from the GC peak area without using a correction factor. The identification of the components was based on a comparison of their mass spectra and linear retention indices (RI, non-polar column), determined with reference to a series of n-alkanes C8–C24 with literature data [42] and computer libraries NIST 2011 and MassFinder 4.1.

### 3.6. Hyaluronidase Inhibition Assay

A hyaluronidase inhibition assay was performed according to the method described by Łyko et al. [13]. Briefly, all extracts and oleanolic acid were diluted in methanol. Then, 20 µL of samples, methanol (blank) or the positive control was mixed with 20 µL of an enzyme (0.1 g/L) and incubated at 37 °C for 10 min. After this time, 20 µL of hyaluronic acid solution (0.5 g/L) was added, and the microplate was incubated for 45 min at 37 °C. For stopping the reaction, 100 µL of acidic albumin solution was added. The transmittance was measured at the wavelength of 600 nm after 10 min of incubation at room temperature. The percentage of hyaluronidase inhibition was calculated according to the following formula:% inhibition = (TS − TBLK)/(TC − TBLK) × 100%,(2)
where TS is the transmittance of the sample (with the addition of the enzyme and substrate), TBLK is the transmittance of the blank with methanol instead of the sample and TC is the transmittance of the negative control (containing the sample, substrate and buffer instead of hyaluronidase solution). At least five concentrations of each sample were examined to plot a dose–response curve and determine the IC_50_ (the concentration of the sample that presents 50% of the maximum inhibition). All measurements were performed in three replicates and averaged.

### 3.7. Lipoxygenase Inhibition Assay (LOX)

The lipoxygenase (LOX) inhibition activity was assayed spectrophotometrically [68]. Briefly, 10 µL of sample was mixed with 240 µL of phosphate buffer (0.066 M; pH 7.8) and 10 µL of enzyme solution (167 U/mL). After the linoleic acid addition (40 µL; 2.5 mM of linoleic acid), the absorbance (λ = 234 nm) was monitored. Then, 50% ethanol instead of the sample was used as a blank, and the quercetin was used as a positive control. All measurements were performed in triplicate. The half-maximal inhibitory concentration (IC_50_) values were calculated based on the dose–response curves, which were created with the use of different concentrations of the extracts and quercetin. The type of inhibition of the enzyme was determined using Lineweaver–Burk plot.

### 3.8. Xanthine Oxidase Inhibition Assay (XO)

The method used for determining the xanthine oxidase (XO) inhibition activity of samples was previously described [76]. Briefly, 40 µL of the extract was mixed with 100 µL of phosphate buffer (0.066 M, pH 7.8) and 20 µL of enzyme (0.01 U/mL). After 10 min of incubation, substrate (115 µL; 0.44 mg/mL of xanthine) was added. The increase in the absorbance was measured (λ = 295 nm) over a 2 min period. Then, 50% ethanol was used instead of the sample as a blank, and the allopurinol was used as a positive control. All measurements were performed in triplicate. IC_50_ values were calculated based on the dose–response curves, created using different concentrations of the extracts. The type of inhibition of the enzyme was determined with the use of the Lineweaver–Burk plot.

### 3.9. Oxygen Radical Absorbance Capacity (ORAC) Assay

The ORAC assay was conducted as described in detail by Olech et al. [77]. Aliquots of 150 μL of fluorescein solution (10 nM) were mixed with 25 μL of sample or Trolox solution (standard) in wells of a black microplate. After 20 min of incubation at 37 °C, 25 μL of AAPH solution (240 mM) was added to each well. The fluorescence was measured after every 90 s for 120 min (excitation wavelength = 485 nm and emission wavelength = 515 nm). A phosphate buffer (75 mM; pH 7.4) instead of the sample was used in a blank solution. Triplicate measurements were performed for each sample and averaged. The results were expressed as Trolox equivalents.

## 4. Conclusions

Our study reports for the first time the terpene and volatiles profile and anti-inflammatory and antioxidant potential of SC-CO_2_ extracts of *Rhododendron luteum* Sweet flowers. As a result of a sequential, two-stage extraction, samples with different degrees of polarity and composition were obtained. The LC-APCI-MS/MS analysis revealed the presence of pentacyclic triterpenes and phytosterol, with the overwhelming majority of oleanolic/ursolic acid being in the polar extract and *β*-sitosterol being in the non-polar extract. The HS-SPME-GC-FID-MS analysis of RLF-CO_2_ showed more than 100 volatile compounds, with benzyl alcohol, *β*-phenylethanol, eugenol and estragole as main components. Many of the metabolites were revealed for the first time for this plant material.

Due to high extraction efficiency, high terpenes recovery and the possibility of being transferred to an industrial scale, the extraction method proposed in this study may be considered an effective method of obtaining bioactive triterpenes and volatiles. Moreover, the proposed SC-CO_2_ extraction protocol uses non-toxic, ecological solvents, matching the trend of “green chemistry”.

Our results clearly indicate that RL lipophilic metabolites (e.g., terpenes) present in the tested samples exhibit anti-inflammatory potential influencing lipoxygenase and xanthine oxidase activity. At the same time, it was concluded that the high ability to inhibit hyaluronidase may be due to the presence of not only terpenes but also other secondary metabolites or some synergistic effects of terpenes and other phytochemicals. Thus, further studies are needed to fully explain the high anti-hyaluronidase activity of the extracts and to identify the compounds responsible for it. In this study, it was also established that RLF SC extracts present an oxygen radical absorbance capacity. Although *Rhododendron luteum* extracts require further analysis, the data presented in this research may be of great importance for the use of RL as a source of active triterpenes and volatiles for potential commercial (e.g., pharmaceutical or cosmeceutical) applications.

## Figures and Tables

**Figure 1 ijms-25-09952-f001:**
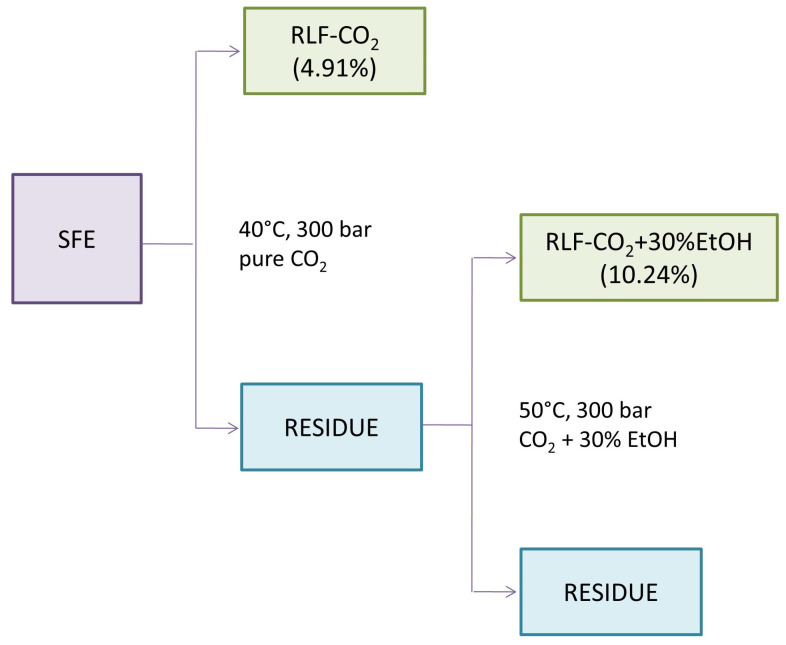
The scheme of the SC-CO_2_ extraction of *Rhododendron luteum* Sweet flowers. In the brackets, the extraction efficiency of specific steps are given. Abbreviations: SFE—Supercritical fluid extraction; RLF-CO_2_—extract obtained with pure CO_2_, RLF-CO_2_ + 30%EtOH—extract obtained with the addition of 30% aqueous ethanol.

**Figure 2 ijms-25-09952-f002:**
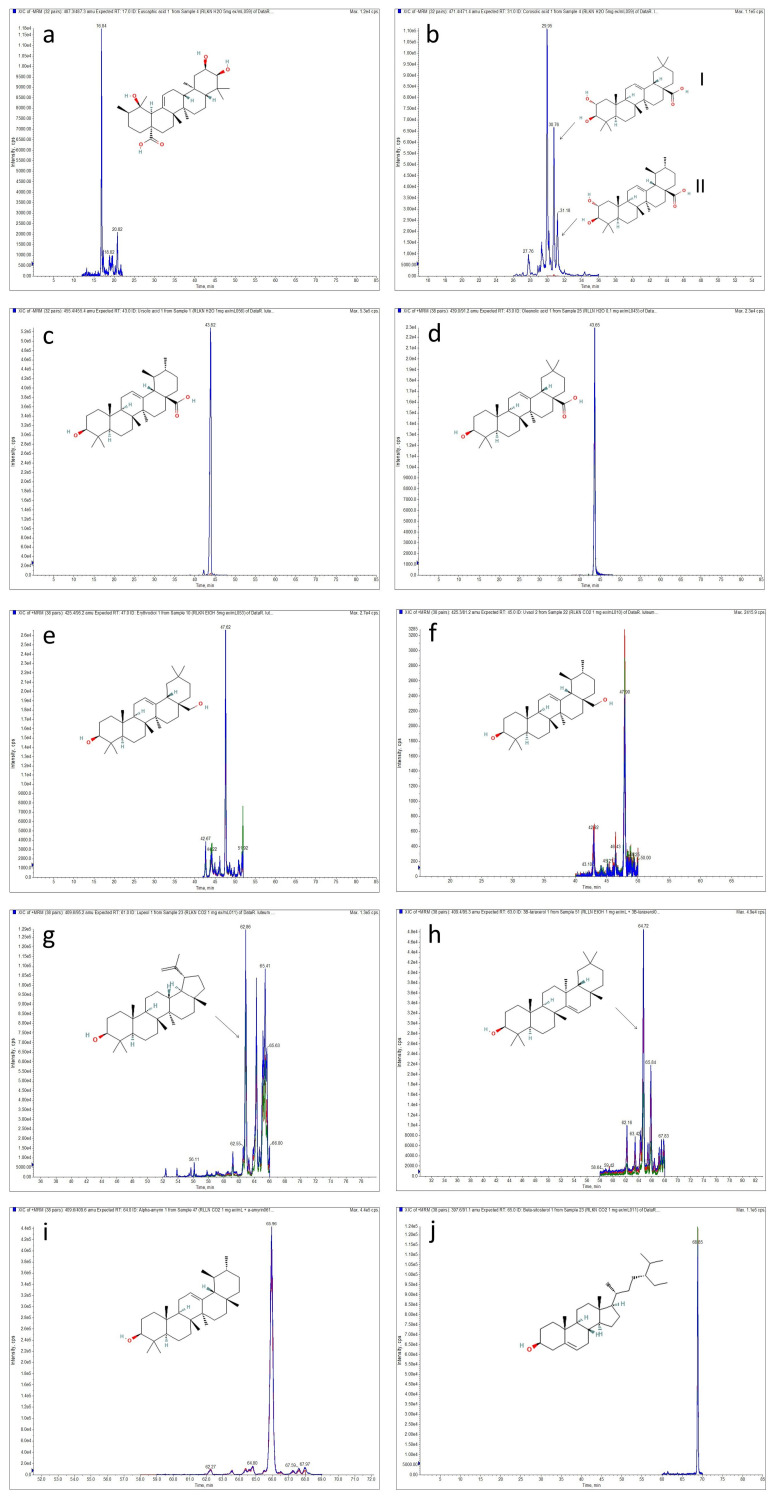
LC-APCI-MS/MS chromatograms obtained in the multiple reaction monitoring (MRM) mode of triterpenes and sterols detected in *Rhododendron luteum* Sweet flowers: (**a**) euscaphic acid; (**b_I_**) maslinic acid; (**b_II_**) corosolic acid; (**c**) ursolic acid; (**d**) oleanolic acid; (**e**) erythrodiol; (**f**) uvaol; (**g**) lupeol; (**h**) 3*β*-taraxerol; (**i**) *α*-amyrin; (**j**) *β*-sitosterol.

**Figure 3 ijms-25-09952-f003:**
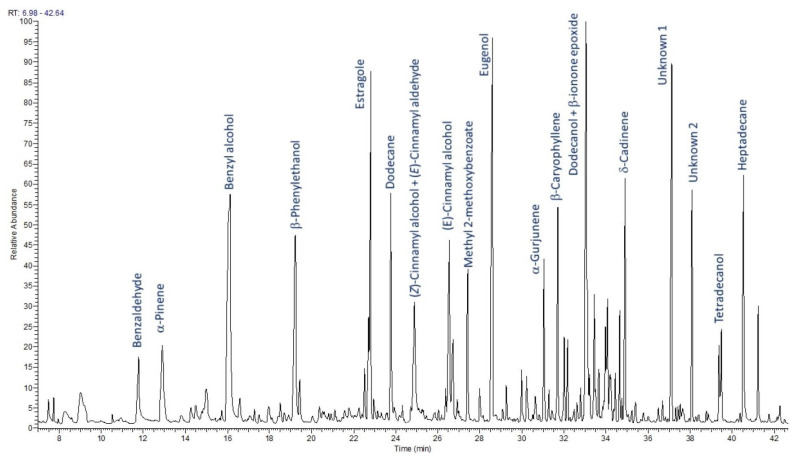
Chromatographic profile of volatile constituents in *Rhododendron luteum* Sweet SFE extract.

**Figure 4 ijms-25-09952-f004:**
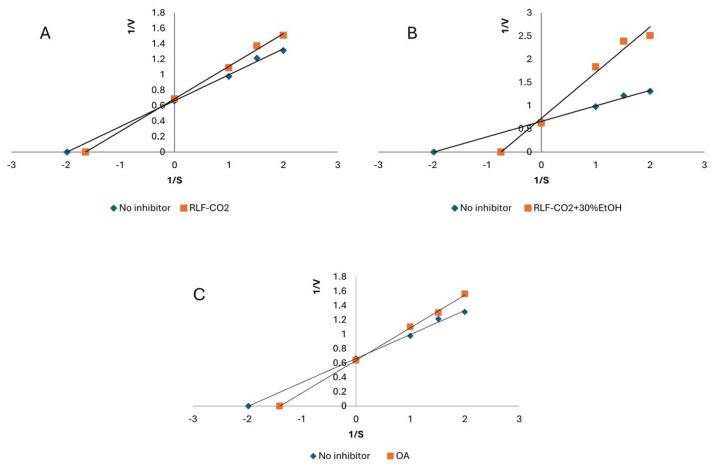
Mode of lipoxygenase (LOX) inhibition by *Rhododendron luteum* flower samples: RLF-CO_2_ (**A**), RLF-CO_2_ + 30%EtOH (**B**) and OA (**C**). Abbreviations are presented as in Table 1.

**Figure 5 ijms-25-09952-f005:**
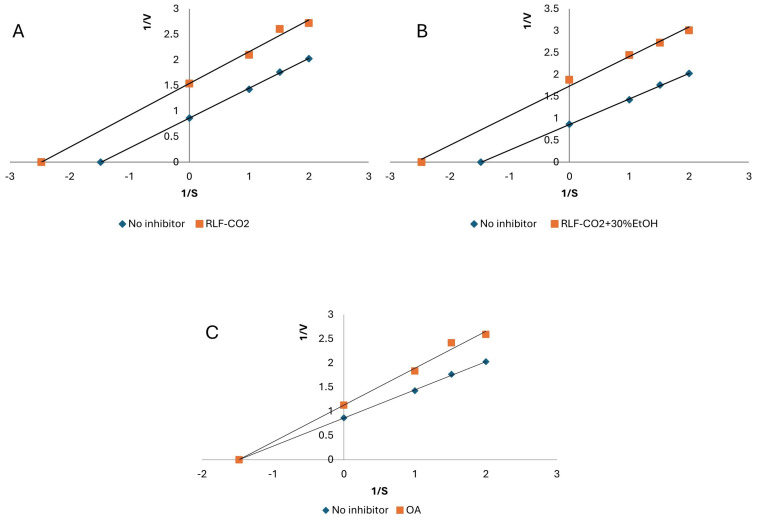
Mode of xanthine oxidase (XO) inhibition by *Rhododendron luteum* flower samples: RLF-CO_2_ (**A**), RLF-CO_2_ + 30%EtOH (**B**) and OA (**C**). Abbreviations are presented as in Table 1.

**Table 1 ijms-25-09952-t001:** Content of triterpenes (mg/g of dry extract (DE)) in *Rhododendron luteum* Sweet flower extracts. Abbreviations: RLF-CO_2_—Extract obtained with pure CO_2_; RLF–CO_2_ + 30%EtOH—extract obtained with the addition of 30% ethanol; BQL—compound detected but below the quantification limit.

	RLF-CO_2_	RLF-CO_2_ + 30%EtOH
*β*-Sitosterol	80.10 ± 0.42	26.87 ± 0.61
3*β*-Taraxerol	49.39 ± 0.05	7.72 ± 0.08
*α*-Amyrin	28.45 ± 0.07	4.81 ± 0.01
Lupeol	17.00 ± 0.14	1.95 ± 0.02
Erythrodiol/Uvaol	3.60 ± 0.03	2.98 ± 0.04
Oleanolic/Ursolic acids	10.43 ± 0.12	94.35 ± 0.35
Corosolic acid	BQL	0.88 ± 0.00
Maslinic acid	0.83 ± 0.01	2.85 ± 0.02
Euscaphic acid	BQL	4.93 ± 0.00

**Table 2 ijms-25-09952-t002:** Volatile constituents identified in *Rhododendron luteum* flower extract.

Constituent	RI_exp_ ^a^	RI_lit_ ^b^	RLF-CO_2_ %
Hexanal	783	785	0.50
Furfural	803	801	t
(*Z*)-Hex-3-en-1-ol	847	851	0.06
Heptanal	882	882	0.10
Benzaldehyde	932	936	0.42
*α*-Pinene	934	936	1.68
6-Methylhept-5-en-2-one	970	972	0.36
*β*-Pinene	973	978	0.35
Hexanoic acid	984	983	0.27
(*E,E*)-Hepta-2,4-dienal	985	987	0.27
Myrcene	986	987	t
Decane	1000	1000	0.34
*α*-Phellandrene	1002	1002	0.19
Benzyl alcohol	1005	1006	10.65
Phenylacetaldehyde	1010	1012	0.50
*p*-Cymene	1013	1015	0.30
*β*-Phellandrene	1020	1023	t
1,8-Cineole	1021	1024	0.10
Limonene	1024	1025	0.42
(*E*)-Oct-2-en-1-al	1033	1034	0.14
(*E*)-*β*-Ocimene	1039	1041	0.27
(*E,Z*)-3,5-Octadien-2-one	1045	1050	0.05
*p*-Cresol	1057	1055	0.19
*trans*-Linalool oxide (f)	1060	1062	0.17
(*E,E*)-octa-3,5-dien-2-one	1068	1070	0.05
Methyl benzoate	1071	1072	0.23
*cis*-Linalool oxide (f)	1074	1072	t
1-Acetyl-2-methylcyclopentene	1080	-	t
Nonanal	1084	1076	0.44
*β*-Phenylethanol	1086	1085	5.20
Benzyl nitrile	1092	1097	0.44
Camphor	1121	1123	0.35
*trans*-Pinocarveol	1125	1126	t
Benzyl acetate	1130	1134	0.10
Menthone	1134	1136	0.07
Ethyl benzoate	1149	1149	0.16
Borneol	1152	1150	0.16
Terpinen-4-ol	1163	1164	0.24
*α*-Terpineol	1174	1176	1.67
Estragole	1176	1179	5.80
Myrtenol	1181	1178	0.35
Decanal	1185	1180	0.10
*β*-Cyclocitral	1196	1195	0.13
Dodecane	1199	1200	5.07
Carvone	1215	1214	0.33
2-Methoxybenzyl alcohol	1227	1223	0.08
(*Z*)-Cinnamyl alcohol	1231	-	2.05
(*E*)-Cinnamaldehyde	1233	1234	0.50
Nonanoic acid	1258	1260	0.20
Thymol	1265	1267	0.28
Bornyl acetate	1270	1270	t
(*E*)-Cinnamyl alcohol	1275	1275	3.77
Carvacrol	1278	1278	1.87
Methyl 2-methoxybenzoate	1297	1295	1.77
Eugenol	1331	1331	7.26
*α*-Cubebene	1350	1355	0.40
Methyleugenol	1365	1369	0.64
*α*-Ylangene	1371	1376	-
*a*-Copaene	1377	1379	0.54
*β*-Bourbonene	1385	1387	0.05
Tetradecane	1400	1400	2.92
*α*-Ionone	1406	1407	0.40
*α*-Gurjunene	1411	1413	0.18
*β*-Caryophyllene	1419	1420	3.09
Geranylacetone	1429	1430	1.16
*β*-Copaene	1430	1403	t
Calarene	1434	1437	0.93
*cis*-Muurola-3,5-diene	1445	1447	0.20
Selina-4(15),6-diene	1447	1449	0.25
*α*-Humulene	1452	1455	0.42
Dodecanol	1459	1460	3.01
*β*-Ionone epoxide	1462	1460	0.71
*β*-Ionone	1465	1467	0.48
*ar*-Curcumene	1472	1473	1.91
*γ*-Curcumene	1473	1474	0.32
Germacrene D	1477	1480	0.65
Eugenol acetate	1485	1483	0.13
*α*-Zingiberene	1487	1489	1.26
Dihydroactinidiolide	1491	1493	1.94
*α*-Selinene	1492	1494	0.44
*α*-Muurolene	1494	1496	0.58
*δ*-Amorphene	1496	1496	0.10
Pentadecane	1550	1500	0.10
*β*-Bisabolene	1501	1503	0.91
*γ*-Cadinene	1507	1507	1.40
*cis*-Calamenene	1510	1517	0.22
*δ*-Cadinene	1515	1520	3.75
Cadina-1,4-diene		1523	0.13
*α*-Calacorene	1530	1527	0.10
*α*-Cadinene	1533	1534	0.44
Selina-3,7(11)-diene	1543	1542	0.08
Spathulenol	1565	1572	0.17
Caryophyllene oxide	1572	1578	0.22
Unknown 1	1583	-	4.23
Ledol	1598	1600	0.19
Hexadecane	1597	1600	0.22
Unknown 2	1616	-	1.88
T-cadinol	1627	1633	0.11
*α*-Cadinol	1639	1642	0.15
Tetradecanol	1659	1661	0.61
Heptadecane	1697	1700	2.25
Benzyl salicylate	1833	1847	0.05
Total identified			89.80

^a^—Experimental retention indices on the nonpolar column; ^b^—literature retention indices according to Adams [42] and MassFinder and NIST libraries; t—trace, <0.04% Unknown 1 (RI 1583), EIMS 70eV, *m*/*z* (%): 161 (100), 93 (75), 43 (47), 123 (41),121 (40), 204 (39), 189 (31), 81 (30), 105 (29), 119 (28), 91 (27), 69 (23), 95 (23), 107 (23), M 222 (2); Unknown 2 (RI 1616), EIMS 70eV, *m*/*z* (%): 161 (100), 43 (59),93 (57),121 (56),119 (42), 163 (37), 105 (32), 91 (29), 81 (27), 120 (27), 107 (26), 95 (25), 189 (24), 69 (22), M 222 (2).

**Table 3 ijms-25-09952-t003:** Anti-inflammatory activity of the main volatile components of *Rhododendron luteum* flower SC-CO_2_ extract.

Compound	Anti-Inflammatory Activity	Reference
Benzyl alcohol	LPS-stimulated inflammatory response suppression through the regulation of NF-κB and AP-1 activity. It is used as an excipient in injectable forms alongside diclofenac sodium. The presence of benzyl alcohol helps enhance the overall anti-inflammatory effects of such formulations.	[49,50]
Eugenol	Inhibiting COX-2, reducing leukocyte migration and rolling in response to chemotactic stimuli. Inhibiting pro-inflammatory mediators, such as IL-1β and TNF-α. Blocking NF-κB activation, which is a key regulator in the inflammatory response. Inhibiting both the COX-2 and 5-LOX pathways.	[45,51]
Estragole	Reducing paw edema induced by carrageenan, dextran, histamine and arachidonic acid. Inhibiting vascular permeability, leukocyte migration and protein extravasation.	[44]
*β*-Caryophyllene	Inhibiting IL-1β-induced nitric oxide (NO) production in human chondrocytes. Reduces the expression of inflammatory mediators such as iNOS, MMP-1 and MMP-13 and activates the CB2 receptor pathway, contributing to its effects in osteoarthritis models.	[52]
(*E*)- and (*Z*)-Cinnamyl alcohol	Inhibition of the lipoxygenase (LOX) pathway, reducing the production of leukotrienes and other pro-inflammatory mediators. In sepsis-induced models, it reduced the levels of pro-inflammatory cytokines IL-1β and IL-18 in the circulatory system by targeting the NLRP3 inflammasome pathway.	[53,54]
Dihydroactinidiolide	Reducing IL-8 production in *Cutibacterium acnes*-stimulated THP-1 cells.	[55]

**Table 4 ijms-25-09952-t004:** Anti-inflammatory and antioxidant activity of *Rhododendron luteum* Sweet flower SFE samples. The results of hyaluronidase, lipoxygenase (LOX) and xanthine oxidase (XO) inhibition are expressed as IC_50_ values. The results of the oxygen radical scavenging capacity (ORAC) are expressed as mg of Trolox/g of dry extract (DE). Abbreviations are presented as in Table 1; OA—oleanolic acid; UA—ursolic acid.

Sample	Hyaluronidase IC_50_ (μg of DE/mL)	LOX IC_50_ (mg of DE/mL)	XO IC_50_ (mg of DE/mL)	ORAC (µg of Trolox/mg of DE)
RLF-CO_2_	106.98 ± 4.45	0.67 ± 0.03	4.50 ± 0.15	119.92 ± 7.33
RLF-CO_2_ + 30%EtOH	23.75 ± 1.44	0.50 ± 0.02	2.36 ± 0.09	338.99 ± 8.6
OA	500.88 ± 10.28	0.68 ± 0,03	2.08 ± 0.09	54.39 ± 2.81
Quercetin	-	9.37 ± 0.00	-	-
Allopurinol	-	-	0.03 ± 0.00	-

## Data Availability

Data are contained within the article or Appendix A.

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
