# Peer review of "Rhododendron luteum* Sweet Flower Supercritical CO_2_ Extracts: Terpenes Composition, Pro-Inflammatory Enzymes Inhibition and Antioxidant Activity"

_ijms, 2024, doi:10.3390/ijms25189952_

Round 1
Reviewer 1 Report
Comments and Suggestions for Authors
This paper employs analytical methods, including LC-APCI-MS/MS and HS-SPME-GC-FID-MS, to thoroughly investigate the chemical composition and anti-inflammatory properties of supercritical carbon dioxide (SCCO2) extract derived from Rhododendron luteum Sweet flowers. The findings significantly contribute to the ongoing development of natural products exhibiting anti-inflammatory effects. The author is encouraged to further delve into the specific mechanisms underlying the extract's action and its anti-inflammatory potential in vivo, establishing a solid foundation for future applied research. For this research, I pose the following questions:
1. Line 56-59 How was the selection of raw materials determined in the study? Given that Rhododendron luteum Sweet is typically toxic, why isn't a non-toxic variety used instead?
2. Line 81-91 Did the researchers compare the components extracted from RL by SFE with those extracted by other methods? What is the difference in the type and content of the extract components, or is there a significant improvement in the activity of the extract?
3. Line 105-190 What are the similarities and differences between the components of extracts from flowers and those from stems and leaves?
4. Line 191-199 The article extensively confirms the advantages of the extraction method, what benefits will this extraction method offer for the industrial production and application prospects?
5. Line 201-236 The study identified over 100 volatile compounds in the extracts. However, the biological activity of these compounds and their overall contributions remain unclear.
6. Line 255-339 The data clearly indicates that the extract exhibits significant inhibition of hyaluronidase, lipoxygenase, and xanthine oxidase activities, but the research lacks a thorough analysis of the mechanisms of action associated with rhododendron extract. However, beyond efficacy assessment, research should also include an evaluation of the extract's safety, such as toxicity and sensitization. This is crucial to confirm its effectiveness within safe usage concentration limits.
7. Line 255-269 The discussion highlighted that certain triterpenes can inhibit hyaluronidase; however, their activity is significantly lower than that of RLF SC CO₂ extracts. It is advisable to compile experimental data for these triterpenes and compare it with their concentrations in the extracts.
8. Line 255-296 Although some triterpenoids, such as oleanolic acid and ursolic acid, have been found in the study, the toxicity data of these compounds have not been mentioned in detail.
9. Line 433-436 To determine the hyaluronidase inhibitory activity of the extract, the article introduces the relevant calculation formulas. However, the annotations regarding TS and TC are not clear. Please clarify what is specifically meant by the term "mixture" in the annotations.
Reviewer 2 Report
Comments and Suggestions for Authors
The research is focused on the evaluation of enzymatic and antioxidant activities of supercritical extracts of Rhododendron luteum flowers. I found the paper engaging, well-structured and accurate, reporting some insights that have not been previously reported. I particularly appreciate the effort done by the authors to investigate the mode of action (competitive, non-competitive,…) of the extracts according to the Lineweaver-Burk plot analysis. As I detected on bibliography, this is the second paper by the same authors, being the first evaluating biological and chemical characterization of the leaf extracts of the same species. However, I would like to introduce some points that I consider that could improve the scope of the manuscript:
- GENERAL ASPECTS:
- Please specify along the paper that the second extraction was done as “RLF-CO2+30%EtOH” (instead of RLF-CO2+EtOH). This could lead to certain misunderstandings.
- ABTRACT:
- I think the abstract is well described, covering all key sections (introduction, methods, results, discussion and conclusion). Besides, the authors exposed an interesting conclusion correlating the biological capacity of the extracts with the triterpene compounds. However, I suggest that the results of the IC50 are not essential to be included in this section, as they are thoroughly detailed in the results section.
- INTRODUCTION:
- L. 69: Authors reported that “the chemical profile of RL flowers is poorly examined, whereby studies of flowers from Eastern European specimens have not been reported to date”. What is known about other regions of the world? Could you provide further explanation regarding researches conducted outside of Eastern Europe?
- L. 87: It is evident that SC offers a promising alternative to other traditional solvent extraction methods. Nonetheless, the cost of using such technology is high, being an evident disadvantage. Could you please discuss not only positive but also negative aspects for SC extraction? Specify how this elevate cost (particularly at large-scale) might impact the profitability on commercializing RL flower extracts?
- RESULTS & DISCUSSION:
- I consider the paragraph from L105 to “particularly lacking” in L108 should be omitted as it is merely described in the introduction. This does not contribute any new results.
- L110: The authors claim that “previous studies” reported… However, no references are provided to support this assertation.
- Figure 1: Is it possible to change the yield of the RLF-CO2 to the upper box and the residue to the lower? I suggest that this adjustment could enhance the clarity of the diagram. This could better reflect the sequence of the two-step extraction: first the RLF-CO2 extraction of the crude flowers (upper box) and then the extraction of the resulting residue (lower box). In addition, the residue was extracted using a 30% hydroalcoholic mixture, please specify also in the chart (as it is somehow confusing)
- Considering that the residue extract (RLF-CO2+EtOH) exhibited superior enzymatic capacities (<IC50) than the crude extract (RLF-CO2), it is suggested that the one-step extract (RLF-CO2+EtOH) of crude flowers could also show outstanding enzymatic capacities. As a result, an important reduction in the economic cost of a two-step extraction should be achieved. Could you consider this possibility? It would be really encouraging to compare the biological activities of the RLF-CO2 and RLF-CO2+EtOH extracts of crude flowers (not the residue).
- L.195: “The proposed two-step supercritical extraction scheme is an effective, environmentally friendly and efficient method”. Indeed, I agree with the authors’ perspective, but the detriment of high extraction costs was not mentioned.
- I have detected a very similar paper using ethyl acetate, methanol and water extracts of this species (https://doi.org/10.1016/j.fct.2019.111052). Could you provide a comparison of the obtained results with those other extracts offering a reasonable discussion on how SC may represent a superior alternative in terms of both biological capacity and profitability?
- MATERIAL & METHODS:
- I do not understand correctly the second extraction. The authors mention that a 38% plant/liquid (30% EtOH) mixture was done and then extracted by SC. Accordingly, was the mixture introduced directly in the reactor and pure CO2 was used as the solvent for the extraction? Why 38% mixture of plant/liquid or 30% EtOH? Is this methodology based on previous tests or studies? Please clarify
- Why is this second extraction more efficient than extracting the residue (30% EtOH) using other extractive methodologies (maceration, mixing, heating,….)?
- L. 379: please change “obtained in a particular step” by “obtained in each particular step”. I think is clearer.
- L. 431: I suppose methanol was also used as blank. Please add it in the description.
Author Response
"Please see the attachment.

Reviewer 3 Report
Comments and Suggestions for Authors
The study carried out is relevant, novel and of interest to the scientific community, in addition, the extraction process by supercritical fluids divided into 2 stages showed interesting results, the level of characterization of the compounds is very detailed and the techniques used are modern and exact.
Some suggestions to improve the manuscript are the following:
1. The materials and methods section can be included before the results and discussion to give clarity and order to the manuscript.
2. I consider that Figure 1 should be placed in the materials and methods section in the subtitle: Supercritical fluid extraction (SFE).
3. Improve the resolution of the images in figure 2, where the chromatograms are shown.
4. It is necessary to cite figure 3, and place it immediately after citing it.
5. Place table 3 in the manuscript once it is cited for the first time.
Round 2
Reviewer 1 Report
Comments and Suggestions for Authors
All the questions are well addressed.
Reviewer 2 Report
Comments and Suggestions for Authors
The authors have considered all the suggestions leading to the improvement of the document. Therefore, I consider that the manuscript should be published in the journal.